# Frequent Users of Emergency Departments: Analysis of the Characteristics and Geographical Distribution in a Local Health Authority in Rome, Italy

**DOI:** 10.3390/healthcare13202609

**Published:** 2025-10-16

**Authors:** Giuseppe Furia, Antonio Vinci, Paolo Lombardo, Paolo Papini, Andrea Barbara, Francesca Mataloni, Mirko Di Martino, Marina Davoli, Massimo Maurici, Gianfranco Damiani, Corrado De Vito

**Affiliations:** 1Department of Public Health and Infectious Diseases, Sapienza University of Rome, 00185 Rome, Italy; 2Local Health Authority Roma 1, Borgo Santo Spirito, 00193 Rome, Italy; 3Department of Biomedicine and Prevention, University of Rome “Tor Vergata”, 00133 Rome, Italymaurici@med.uniroma2.it (M.M.); 4Department of Epidemiology, Lazio Regional Health Service, 00147 Rome, Italy; 5Department of Life Sciences and Public Health, Università Cattolica del Sacro Cuore, Largo Francesco Vito 1, 00168 Rome, Italy; gianfranco.damiani@unicatt.it; 6Department of Woman and Child Health and Public Health, Fondazione Policlinico Universitario “A. Gemelli” IRCCS, Largo Agostino Gemelli 8, 00168 Rome, Italy

**Keywords:** Emergency Departments, Rome, Italy, health equity, social determinants of health

## Abstract

**Background/Objectives:** Frequent users (FUs) are patients who repeatedly attend Emergency Departments (EDs). This study aims to identify the clinical and social characteristics of FUs in a Local Health Authority in Rome and to quantify and compare the variation in the probability of being FU attributable to General Practitioners (GPs) and Local Health Districts (LHDs). **Methods:** The Healthcare Emergency Information System and an automated database of Lazio Region residents were used for the collection of data on the patients’ socioeconomic status, GP, LHD and chronic diseases. Different FU thresholds (attendances ≥4, 5, 7 or 10) were used for descriptive analyses. Univariate logistic analysis and a multilevel logistic model were performed for inferential analyses. **Results:** A total of 89,036 individuals attended at least one of the 13 EDs included in the study. Mental illness was present in 2.6% of non-FUs compared with 7.6% of FUs with ≥4 attendances. The OR of being FU increased with higher clinical complexity. GP appeared to play an important role in determining FU behavior, while no significant effect was found on the LHD level. **Conclusions:** This study identified potential risk factors predictive of disproportionate ED use and may help policymakers address the FU phenomenon.

## 1. Introduction

The Italian population, like other OECD countries, has experienced an increase in chronic illnesses over the past few decades, with population aging being one of the primary drivers [1]. Despite significant improvements in the emergency care system, the growing number of frail patients and their care needs have led to a rise in Emergency Department (ED) attendance [2,3]. In hospitals with limited inpatient bed capacity, increasing ED attendances and the number of emergency patients requiring admission may increase the length of stay in the ED, leading to overcrowding and compromised ED performance, as well as to the increase of inappropriate hospitalizations and health care costs. Reducing ED utilization has therefore become a crucial goal for new healthcare delivery models [4,5,6]. However, the predictors of ED attendance are complex and multifactorial [7].

There is no single definition of Frequent Users (FUs), and it depends on the length of observation time. Frequent Users are patients with at least four ED attendances per year, as defined in most of the literature [8,9,10,11,12,13]. Although they account for only 4.5–8% of ED patients, they are responsible for up to 28% of all ED attendances [10,14]. Excessive ED use raises two major concerns: the first is associated with additional ED costs, which often exceed the government’s budgetary limits, particularly since the end of the COVID-19 pandemic [15,16,17]; the second is compromising the quality of services provided to patients with acute conditions requiring appropriate ED attendance. Furthermore, it contributes to overcrowding and increases the likelihood of medical errors [8,9,18].

Previous research has highlighted that FUs are more likely to have chronic conditions (particularly renal failure, cardiovascular disease, diabetes and chronic pulmonary disease), mental illness or substance use disorders [19,20]. The prevalence of chronic conditions is higher among FUs than in the general population, and timely interventions in primary care could help prevent ED attendances and hospitalization [21,22]. A growing body of evidence reveals that FUs frequently use other healthcare services (other than the ED), are more often of lower socioeconomic status, and experience health outcomes strongly influenced by their residential context [19,21,22,23,24,25,26,27].

As in other countries, Italy has recently focused on improving care coordination and reducing ED attendance and hospital admissions by promoting more appropriate use of community-based services [28]. Compared with other European contexts, a relatively high proportion of Italians report that they have unmet medical needs, and access to care is particularly difficult for those of lower socioeconomic status [29].

Analyzing the socioeconomic profile and the geographic variation in ED use may therefore be an essential first step in guiding local policies in reducing avoidable emergency department attendances in Italy [30].

According to the literature, the prevalence of health problems is a key determinant in ED attendance frequency. Nevertheless, other factors have also been associated with high levels of ED attendance [31,32,33,34], especially in urban peripheries and inner-city areas, such as housing and employment problems, loneliness and marginalization, limited social support, proximity to the ED, and poor access to primary care [35,36,37].

Moving from macro-level analyses to more granular units such as census tracts makes it possible to identify local hotspots of high healthcare utilization and to highlight contextual factors affecting specific populations and communities. This approach provides insight into how interventions can be geographically targeted to improve healthcare delivery [38].

The aim of this study is to identify the clinical and social characteristics of FUs and to quantify and compare the variation in the probability of being FU attributable to GPs and Local Health Districts (LHDs), while also assessing socio-economic status and chronic conditions. The findings obtained by this combination of both individual and population level data may support policymakers and healthcare providers in developing care coordination programs and strategies, with the ultimate goal of addressing social and territorial inequalities in healthcare access, reducing avoidable ED usage, and encouraging a more appropriate usage of preventative and primary care opportunities.

## 2. Materials and Methods

### 2.1. Study Design

This study was carried out in Local Health Authority (LHA) Roma 1 (see Appendix A), one of the three LHAs in the municipality of Rome, in the Lazio Region. It comprises six Local Health Districts (LHD 1, LHD 2, LHD 3, LHD 13, LHD 14, LHD 15) and 805 General Practitioners (GPs), and serves approximately one million residents.

While LHA Roma 1 is not fully representative of the entire Italian or Regional setting, it is one of the most populous areas in Italy, with 13 EDs (out of the 22 in the Rome metropolitan area), an aging index (population aged 65+ divided by those aged 0–14, per 100) of 202 in 2024 (compared with a regional mean of 189) and an old-age dependency ratio (population aged 65+ divided by population in the working age 15–64) of 38.1 in 2024 [39]. This retrospective cohort study included all ED attendances from 1 January 2021, to 31 December 2021, in all 13 Emergency Departments within LHA Roma 1. All patients assisted by a GP from the same LHA were enrolled in the cohort. Patients younger than 18 years and those who attended single-specialty EDs (ophthalmology, pediatrics) and attendances for obstetric or gynecological problems were excluded since they represent a population with specific needs that is worthy of specific focus and not comparable with the general population.

FUs were defined as patients with ≥4 ED attendances per year Given the absence of a universal definition, different cut-offs (attendances ≥5, 7 or 10) were also considered in the descriptive analysis.

For each patient in the cohort the following potential risk factors were assessed: gender, age, socioeconomic status (high, middle-high, medium, middle-low, low) and the presence of chronic or multiple-chronic conditions [40,41]. The socioeconomic level was calculated at the census tract level, based on the methodology developed by Nicola Caranci and colleagues [42]. This index integrates multiple socio-economic indicators derived from national census data, including educational attainment, employment status, home ownership versus rental, household overcrowding, and family structure. It provides a composite measure of socio-economic disadvantage within small geographic areas. Among patients with multiple chronic conditions, high clinical complexity was defined as a five-year mortality risk higher than 10%, based on the number and type of chronic diseases [43].

Data related to ED attendances were also compared to identify differences between FUs and non-FUs. In particular, triage score, principal diagnosis group in the ED, and reported symptoms at the time of arrival were considered.

### 2.2. Data Sources

The cohort was derived from the Healthcare Emergency Information System, which collects all attendances to emergency services, patient demographic data, visit and discharge dates and times, ICD-9-CM diagnosis, reported symptoms on attendance, triage score (from no urgent to emergency), and discharge status (e.g., dead, hospitalized, or discharged at home).

The cohort was linked to the automated databases of Lazio Region residents who receive NHS assistance, thus allowing researchers to obtain information related to chronic or multiple chronic diseases, GP and LHD of each patient, and socioeconomic status based on the residence address [42,44,45]. A deterministic record linkage procedure with anonymous identification codes was used to merge the data from different information systems. To preserve privacy, each individual identification code was subsequently and automatically deidentified, and the conversion table was deleted, leaving only fully anonymized data available to researchers.

### 2.3. Geographical Analysis

The administrative-territorial division of the LHA Roma 1 was used to examine the association between FU prevalence and urban settings, as previously described [46]. Each of the six LHDs of LHA Roma 1 is divided into Geographical Units (GUs, in Italy called “Zone Urbanistiche”), as defined by the Municipality of Rome. This represent the smallest territorial unit for which population data are available in Italy and many other countries [47,48].

### 2.4. Statistical Analysis

The outcome variable was binary (FU ≥ 4 vs. non-FU). Frequency distributions of FUs (attendances ≥ 4, 5, 7 and 10) by triage, principal diagnosis and presenting symptoms at ED arrival were compared with those of non-FU. A descriptive analysis was performed reporting absolute frequencies of patient groups and ED attendances characteristics. Univariate logistic analysis was used to identify candidate predictor variables among those collected in the dataset. A multilevel logistic model (patient < GP < District) was performed to quantify the variability in ED FU behavior attributable to LHDs and primary care physicians and to identify the role of social and clinical determinants (gender, age, socioeconomic level and chronic conditions).

Age was considered as a continuous variable. The Box-Tidwell test was used to check for linearity between the “predictor” and the logit. The effect of individual variables was expressed as Odds Ratios (OR); variance components estimated by multilevel models were expressed as Median Odds Ratios (MORs). The MOR quantifies between-cluster variation by comparing two patients from two randomly chosen, different clusters. Consider two persons with the same covariates, chosen randomly from two different clusters. The MOR represents the median odds ratio between the person of higher propensity and the person of lower propensity. MOR values are always ≥1.00; if MOR = 1.00, there is no variation between clusters, while larger values indicate greater variation [49]. MORs were estimated for both the “empty” model, which includes a random intercept only, and the full model, which includes all patient risk factors. Statistical analyses were conducted using SAS software 9.4 version (SAS Institute Inc., Cary, NC, USA).

## 3. Results

### 3.1. Descriptive Analysis

A total of 89,036 patients accessed at least one of the 13 EDs in LHA Roma 1 during 2021. Patients and ED attendances characteristics are summarized in Table 1 and Table 2. A total of 72,781 patients were registered with a GP in LHA Roma 1. Overall, patients accounted for 99.811 attendances. FUs represented a small fraction of the overall population (2.7%) but were responsible for a large share of attendance (11.3%). Among non-FUs, females accounted for 51.9%, while males were more frequent among FUs (ranging from 51.9% in FU ≥ 5 to 57.7% in FU ≥ 10) than among non-FUs (48.1%). The 50–59 age group was the most frequent among non-FUs (18.4%) and FU ≥ 4 (17.5%), and its proportion increased with the number of FU attendances. Low socioeconomic status was prevalent among most FUs (from 25.3% of FU ≥ 4 to 24.6% of FU ≥ 10), whereas high and medium-high levels were more frequent among non-FUs. Multiple chronic conditions were present in 29.4% of FU ≥ 4 patients, whereas 47.6% of FU ≥ 4 patients had no chronic disorders recorded.

Table 2 was constructed after classifying patients according to the number of ED attendances. Emergency and urgent triage codes decreased from 24.9% in the FU ≥ 4 group to 20.9% in the FU ≥ 10 group. Conversely, non-urgent codes increased from 8.1% in the FU ≥ 4 group to 14.4% in the FU ≥ 10 group. The most frequent diagnoses among all patients fell into the category of “Symptoms, signs and ill-defined conditions”. Mental disorders were recorded in 2.6% of the non-FU group but increased from 7.6% in the FU ≥ 4 group to 15.9% in the FU ≥ 10 group. Cardiovascular diseases were similar in the non-FU and FU ≥ 4 groups (8.6% and 8.2%, respectively). Excluding nonspecific and missing diagnoses, many acute conditions (e.g., trauma, abdominal and chest pain, dyspnea) decreased in frequency from the non-FU group to the FU ≥ 10 group. In contrast, psychomotor agitation increased from 5.1% in the FU ≥ 4 group to 11.2% in the FU ≥ 10 group. Social issues were reported in 0.4% of diagnoses in the FU ≥ 4 group and 0.9% in the FU ≥ 10 group, whereas none were reported in the non-FU group.

### 3.2. Geographical Distribution

The geographical classification included two levels: GUs and LHDs. For each level, five classes were defined based on the relative percentage of FUs in the area. The geographical distribution shown in Figure 1 indicates that some areas of LHA Roma 1 had a significantly higher percentage of FUs, particularly in LHD 1, LHD 14 and LHD 15.

### 3.3. Inferential Analysis

Results of the multilevel analysis are reported in Table 3. Among candidate predictor variables identified in the descriptive and geographical analyses, the strongest effect was associated with the presence of multiple chronic conditions: the greater the patient’s clinical complexity, the higher the OR of being a FU. OR increased with decreasing socioeconomic status, with a more pronounced effect for low and medium-low levels and a less consistent effect for higher levels. The role of GP did not show a consistent influence on FU behavior (MOR 1.18, Wald *p* = 0.061), and no significant effect was observed at the LHD level (MOR 1.05, Wald *p* = 0.127).

## 4. Discussion

FUs represented 2.7% of the overall population but were accounted for 11.3% of attendances in 2021, in line with the literature [46], most frequently in the 50–59 age group. The risk of being FU increased with the patient’s clinical complexity and with low and medium-low socioeconomic status. Generally, patients with higher socioeconomic status were less likely to attend the ED. This is consistent with a study conducted in Milan, where the odds of avoidable hospital admissions were higher among patients with lower socioeconomic status compared with other groups [50]. Prior studies suggest that patients with low socioeconomic level perceive ED assistance to be cheaper and more accessible than ambulatory care and are often more likely to use EDs for nonurgent conditions [33,49,51].

In this study, non-urgent triage codes were more frequent among FUs (from 8.1% in the FU ≥ 4 group to 14.4% in the FU ≥ 10 group), and mental disorders were common in a substantial proportion of FUs (to 15.9% in the FU ≥ 10 group). Psychomotor agitation and social issues were important diagnoses associated with FUs, but the results of “symptoms, signs and ill-defined conditions” and “external causes of injury and supplemental classification” diagnosis groups, as well as the main issues on admission, such as fever, chest pain or dyspnea, may have been influenced by the COVID-19 pandemic. This may also have affected time-dependent conditions such as stroke or cardiac complaints, possibly due to concerns about acquiring COVID-19 in hospital [52,53]. However, all patients included in the analysis (those with at least one ED visit in 2021) were equally exposed to the pandemic waves, and no systematic differences were expected between subgroups that could have introduced bias. Furthermore, other studies in the Lazio Region comparing ED attendances during the COVID-19 waves found a sharp reduction in ED attendances, except for pneumonia [54]. The increase of physical and mental morbidities was associated with higher ED attendance rates, supporting previous evidence that individuals with chronic diseases and psychiatric disorders are more likely to attend EDs [19,55]. This also applies to individuals with socioeconomic deprivation [31,56]. Across the EU, admissions related to chronic and psychiatric disorders are estimated to be potentially avoidable through improved prevention and disease management, which could reduce reliance on EDs [57,58]. Some authors have investigated the importance of social support for older adults in the ED, although in a systematic review there was no significant association between ED attendance and social support [59].

Regarding the geographical analysis shown in Figure 1, this study characterized variation using more granular geographic unit of analysis. This may be considered a first step toward developing specific public health strategies aimed at improving appropriate healthcare utilization for specific populations and communities, as implemented in Paris by the Île-de-France Regional Health Agency [46]. In other studies on geographic variation in healthcare, utilization and outcomes were quantified by using geographic macro-areas, but it was impossible to distinguish a single neighborhood characterized by different socioeconomic and demographic characteristics [30,50,60,61]. Further studies are needed to analyze the association among FUs, population density, income and medical services or GP offices in the same area.

The multilevel analysis indicated that GPs did not play a consistent role in avoiding FU behaviors (MOR 1.18, Wald *p* = 0.061), similar to findings at the territorial level (LHDs). It is possible that FUs bypass GPs to obtain quicker treatment. In some studies, reductions in ED demand were associated with the availability of at-home GP visits or improved public transportation [62]. Other authors have reported that frequent ED users also have high rates of GP consultations and outpatient care in addition to ED use. Therefore, efforts to simply increase access to primary care or extend GP opening hours (weekdays and weekends) may not necessarily reduce ED utilization [63].

Primary prevention of the FU phenomenon has been little discussed in the literature. These FU subpopulations may represent distinct targets for interventions, such as tailored individual action plans [64]. Individual care plans for FUs are often based on secondary prevention, initiated after an ED attendance and followed by intervention or case management [65]. Policies should focus on comprehensive patient and family care, considering hospital and territorial services, consistently combining medical and social dimensions [66].

Coordinated, team-based approaches integrating medical, behavioral, and care management services have been recognized as a cost-effective model to reduce ED attendances and combined ED and inpatient hospital costs among patients with complex care needs, even those who report access to traditional primary care. The case management approach is a tool that can be applied in different ways and contexts [66,67].

Two systematic reviews showed that compared with standardized methods, a customized case management approach helps FUs find appropriate responses to their needs. The main tool used is the individual care plan with phone contact, supportive group therapy and electronic systems for rapid patient identification. Teams are heterogeneous, although nurses are often the most frequently involved professionals [68].

High rates of FU-related ED attendances may also reflect poverty in metropolitan areas and highlight social inequalities in healthcare access. As emphasized in the New Urban Agenda, the United Nations Sustainable Development Goals promote health through several interconnected health-related targets, achievable via multisectoral approaches [69]. Thus, reducing FU rates and improving the health status of the general population may be achieved through joint policies with other partners, such as schools or transportation agencies [70,71,72].

### Limitations and Strengths

This study represents the most comprehensive analysis in a single metropolitan area in Italy including all FU attendances over one year and considering their health status, socioeconomic determinants, and geographical distribution. Although these findings are not generalizable to the entire population, they are relevant to many urban areas with similar levels of social deprivation. Therefore, institutions and healthcare providers may be able to identify territories where residents are at higher risk of developing FU patterns and to suggest primary prevention measures. Access to regional data flows minimized the risk of missing records or information. Moreover, residents living in extremely disadvantaged circumstances were likely captured in our data due to the fictitious address assigned to them by the LHA.

This study has some limitations. The retrospective design allowed investigation of the predictors of ED attendance only at a single point in time and exclusively among LHA residents, thereby excluding homeless individuals without residency, foreigners, and people formally resident in LHA Roma 1 but whose healthcare services were provided by other LHAs in Rome. Only the main diagnosis was considered, and inaccuracies in the clinical dataset may have led to underreporting of some morbidities. The potential influence of proximity to EDs and primary care services on ED attendance was not evaluated. Additional variables would be necessary to quantify COVID-19-specific attendance, to better understand their influence on the results.

## 5. Conclusions

The results might not be generalizable to the Italian context, but they are extremely relevant to metropolitan areas with similar socio-demographic composition, as is the case for many major Italian and European cities.

Frequent ED use represents a major challenge for healthcare system management. Analysis of ED attendances and the socioeconomic and geographical characteristics of FUs highlights the need for new approaches to address key issues such as socioeconomic inequalities, improvement of housing and employment conditions, and structural factors including the strategic placement of primary care services and improved transportation.

This study enabled the identification of potential risk factors predictive of disproportionate ED use and supports policymakers in anticipating the needs of specific patient groups or categories. Further studies are warranted to examine the prevalence of FUs and the geographical distribution of hospitals, residents’ income and primary care services across the entire Lazio Region. It would also be useful to investigate the effectiveness of territorial interventions, in line with the new directives of the National Recovery and Resilience Plan for chronic disease management at the territorial level.

## Figures and Tables

**Figure 1 healthcare-13-02609-f001:**
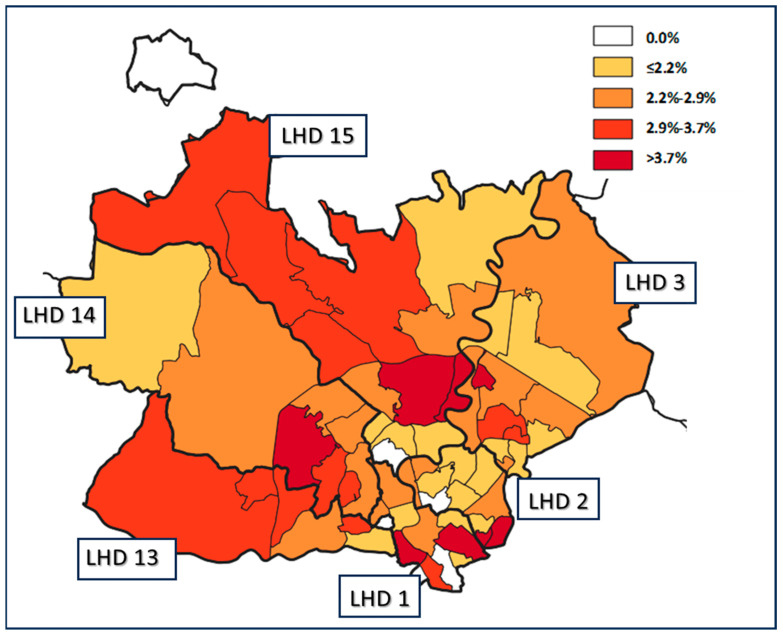
Geographical distribution of FUs. Colors represent the five percentage cut-off groups of FUs at the Geographical Unit population level. The bold black line delineates the six LHDs, while the thin black line delineates the Geographical Units.

**Table 1 healthcare-13-02609-t001:** Characteristics of FU and non-FU patients in 2021.

		Patients		Non-FU		FU ≥ 4		FU ≥ 5		FU ≥ 7		FU ≥ 10	
		N	%	N	%	N	%	N	%	N	%	N	%
Total		72,781	100.0	70,743	100.0	2038	100.0	977	100.0	349	100.0	130	100.0
Gender	Male	35,123	48.3	34,054	48.1	1069	52.5	507	51.9	187	53.6	75	57.7
	Female	37,658	51.7	36,689	51.9	969	47.5	470	48.1	162	46.4	55	42.3
Age	18–29	10,104	13.9	9885	14.0	219	10.7	99	10.1	35	10.0	14	10.8
	30–39	7484	10.3	7285	10.3	199	9.8	96	9.8	42	12.0	18	13.8
	40–49	10,512	14.4	10,243	14.5	269	13.2	133	13.6	55	15.8	21	16.2
	50–59	13,381	18.4	13,024	18.4	357	17.5	181	18.5	70	20.1	34	26.2
	60–69	10,235	14.1	9933	14.0	302	14.8	147	15.0	57	16.3	21	16.2
	70–79	10,066	13.8	9739	13.8	327	16.0	147	15.0	42	12.0	10	7.7
	80+	10,999	15.1	10,634	15.0	365	17.9	174	17.8	48	13.8	12	9.2
Socio-economic level	High	16,680	22.9	16,297	23.0	383	18.8	177	18.1	66	18.9	26	20.0
	Medium-high	17,916	24.6	17,473	24.7	443	21.7	206	21.1	73	20.9	28	21.5
	Medium	13,398	18.4	13,051	18.4	347	17.0	161	16.5	52	14.9	20	15.4
	Medium-low	11,695	16.1	11,346	16.0	349	17.1	165	16.9	53	15.2	24	18.5
	Low	13,092	18.0	12,576	17.8	516	25.3	268	27.4	105	30.1	32	24.6
Chronic conditions	No chronic conditions	43,531	59.8	42,561	60.2	970	47.6	451	46.2	156	44.7	59	45.4
	One chronic condition	15,718	21.6	15,249	21.6	469	23.0	226	23.1	83	23.8	40	30.8
	Multiple chronic conditions (low-mid clinical complexity)	9345	12.8	9035	12.8	310	15.2	144	14.7	45	12.9	9	6.9
	Multiple chronic conditions (high clinical complexity)	4187	5.8	3898	5.5	289	14.2	156	16.0	65	18.6	22	16.9

**Table 2 healthcare-13-02609-t002:** Characteristics of ED attendances of FU and non-FU patients in 2021.

		Attendances		Non-FU		FU ≥ 4		FU ≥ 5		FU ≥ 7		FU ≥ 10	
		N	%	N	%	N	%	N	%	N	%	N	%
Total		99,811	100.0	88,514	100.0	11,297	100.0	7053	100.0	3721	100.0	2046	100.0
Triage admission code	Emergency	5150	5.2	4528	5.1	622	5.5	345	4.9	175	4.7	95	4.6
	Urgency	19,176	19.2	16,986	19.2	2190	19.4	1327	18.8	651	17.5	334	16.3
	Deferrable Urgency	35,505	35.6	31,605	35.7	3900	34.5	2269	32.2	1157	31.1	610	29.8
	Minor urgency	36,530	36.6	32,861	37.1	3669	32.5	2389	33.9	1208	32.5	712	34.8
	Non urgency	3450	3.5	2534	2.9	916	8.1	723	10.3	530	14.2	295	14.4
ICD-9-CM Diagnosis group on discharge	Infectious and parasitic diseases	1625	1.6	1469	1.7	156	1.4	92	1.3	40	1.1	21	1.0
	Neoplasms	675	0.7	532	0.6	143	1.3	86	1.2	39	1.0	20	1.0
	Endocrine, nutritional and metabolic diseases, and immunity disorders	686	0.7	580	0.7	106	0.9	63	0.9	29	0.8	11	0.5
	Diseases of the blood and blood-forming organs	1150	1.2	894	1.0	256	2.3	184	2.6	96	2.6	54	2.6
	Mental disorders	3173	3.2	2315	2.6	858	7.6	656	9.3	444	11.9	325	15.9
	Diseases of nervous system and sense organs	3596	3.6	3165	3.6	431	3.8	268	3.8	139	3.7	66	3.2
	Disease of the circulatory system	8514	8.5	7592	8.6	922	8.2	478	6.8	197	5.3	92	4.5
	Diseases of the respiratory system	4136	4.1	3740	4.2	396	3.5	215	3.0	107	2.9	28	1.4
	Diseases of the digestive system	6866	6.9	6110	6.9	756	6.7	443	6.3	192	5.2	85	4.2
	Diseases of the genitourinary system	2905	2.9	2450	2.8	455	4.0	291	4.1	98	2.6	38	1.9
	Complications of pregnancy, childbirth and puerperium	675	0.7	622	0.7	53	0.5	34	0.5	5	0.1	3	0.1
	Diseases of the skin and subcutaneous tissue	961	1.0	876	1.0	85	0.8	39	0.6	19	0.5	6	0.3
	Diseases of the musculoskeletal system and connective tissue	6153	6.2	5651	6.4	502	4.4	262	3.7	127	3.4	63	3.1
	Congenital anomalies	566	0.6	495	0.6	71	0.6	47	0.7	20	0.5	12	0.6
	Certain conditions originating in the perinatal period	46	0.0	41	0.0	5	0.0	2	0.0	2	0.1	2	0.1
	Symptoms, signs and ill-defined conditions	21,504	21.5	18,781	21.2	2723	24.1	1713	24.3	856	23.0	425	20.8
	Injury and poisoning	27,269	27.3	25,818	29.2	1451	12.8	745	10.6	340	9.1	192	9.4
	External causes of injury and supplemental classification	2601	2.6	1770	2.0	831	7.4	669	9.5	485	13.0	258	12.6
	Missing	6710	6.7	5613	6.3	1097	9.7	766	10.9	486	13.1	345	16.9
Main issue on admission	Coma	13	0.0	10	0.0	3	0.0	1	0.0	1	0.0	1	0.0
	Shock	12	0.0	12	0.0	0	0.0	0	0.0	0	0.0	0	0.0
	Dyspnea	3626	3.6	3222	3.6	404	3.6	221	3.1	107	2.9	40	2.0
	Abdominal Pain	8087	8.1	7053	8.0	1034	9.2	674	9.6	347	9.3	190	9.3
	Neck Pain	164	0.2	145	0.2	19	0.2	9	0.1	1	0.0	0	0.0
	Chest Pain	5362	5.4	4784	5.4	578	5.1	319	4.5	168	4.5	84	4.1
	Non-traumatic bleeding	1093	1.1	953	1.1	140	1.2	81	1.1	33	0.9	8	0.4
	Fever	3206	3.2	2895	3.3	311	2.8	186	2.6	62	1.7	23	1.1
	Intoxication	288	0.3	214	0.2	74	0.7	52	0.7	35	0.9	32	1.6
	Hypertension	1122	1.1	996	1.1	126	1.1	75	1.1	37	1.0	18	0.9
	Rhythm alteration	1829	1.8	1642	1.9	187	1.7	102	1.4	32	0.9	14	0.7
	Acute neurological syndrome	871	0.9	799	0.9	72	0.6	42	0.6	16	0.4	8	0.4
	Other nervous system symptoms	2236	2.2	2010	2.3	226	2.0	133	1.9	77	2.1	43	2.1
	Social issues	51	0.1	20	0.0	31	0.3	27	0.4	22	0.6	18	0.9
	Medico-legal checks	53	0.1	38	0.0	15	0.1	11	0.2	9	0.2	7	0.3
	Allergic reaction	450	0.5	419	0.5	31	0.3	13	0.2	7	0.2	4	0.2
	Trauma or burn	28,670	28.7	27,344	30.9	1326	11.7	680	9.6	298	8.0	167	8.2
	Dermatological disorders	313	0.3	286	0.3	27	0.2	12	0.2	7	0.2	4	0.2
	Eye symptoms or disorders	726	0.7	541	0.6	185	1.6	130	1.8	78	2.1	37	1.8
	Dental disorders	2024	2.0	1795	2.0	229	2.0	115	1.6	44	1.2	13	0.6
	ENT symptoms or disorders	1108	1.1	980	1.1	128	1.1	70	1.0	28	0.8	10	0.5
	Urological symptoms or disorders	1860	1.9	1452	1.6	408	3.6	256	3.6	96	2.6	42	2.1
	Psychomotor agitation	1594	1.6	1021	1.2	573	5.1	438	6.2	294	7.9	229	11.2
	Other	35,048	35.1	29,879	33.8	5169	45.8	3406	48.3	1922	51.7	1054	51.5

**Table 3 healthcare-13-02609-t003:** Multilevel logistic regression model predicting ED frequent usage.

Predictor Variable			OR	95% Confidence Limits		*p* Value
Age (years)			1.00	1.00	1.00	0.794
Gender	Male		reference	-	-	-
	Female		0.87	0.79	0.95	0.002
Chronic conditions	No chronic conditions		reference	-	-	-
	One chronic condition		1.37	1.20	1.56	<0.001
	Multiple chronic conditions (low-mid clinical complexity)		1.54	1.32	1.80	<0.001
	Multiple chronic conditions (high clinical complexity)		3.18	2.70	3.76	<0.001
Socio-economic level	High		reference	-	-	-
	Medium-high		1.07	0.92	1.25	0.335
	Medium		1.12	0.95	1.31	0.166
	Medium-low		1.29	1.09	1.51	0.004
	Low		1.70	1.46	1.97	<0.001
**Multilevel parameters**						
**Intercept-only model**		**MOR**			***p*** **(wald)**	
LHD *		1.08			0.127	
GP *		1.24			0.008	
**Full model**		**MOR**			***p*** **(wald)**	
LHD		1.05			0.207	
GP		1.18			0.061	

* LHD: Local Health District; GP: General Practitioner.

## Data Availability

The datasets used and analyzed during the current study are available from the corresponding author on reasonable request.

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
