# Peer review of "Frequent Users of Emergency Departments: Analysis of the Characteristics and Geographical Distribution in a Local Health Authority in Rome, Italy"

_healthcare, 2025, doi:10.3390/healthcare13202609_

Round 1
Reviewer 1 Report
Comments and Suggestions for Authors
Thank you for the interesting read. Despite this fact, the paper presents some limitations. in terms of scope and generalizability, the study is restricted to a single year (2021), a period strongly influenced by COVID-19, which the authors acknowledge but do not model explicitly. Thus, many findings (declines in acute presentations, increases in psychiatric/social conditions) may reflect pandemic anomalies. Also, limited to one LHA in Rome – while populous, extrapolation to other Italian or EU contexts requires caution.
Regarding the role of GPs and primary care, the result that GP influence is minimal is surprising given Italian reliance on family doctors. Yet, the analysis only considers GPs as statistical clusters, without adjusting for GP-level characteristics (age, practice size, urban/rural differences, organizational model). This weakens the interpretation that GPs are not influential.
We also believe that the paper has an underdeveloped discussion. The paper notes similarities with Milan and Paris but could better contextualize how Italian structural issues (GP gatekeeping role, resource shortages, transport) differ from Northern European models. Also, the discussion of intervention strategies remains surface-level. Case management, integration of mental health/social services, and digital solutions are only briefly hinted at but not fully explored.
So, taking all the above, to strengthen the paper, the following revisions are recommended:
-
Clarify the FU definition rationale: justify why ≥ 4 was chosen over ≥3 or ≥5. Possibly add sensitivity analysis of risk factors across cut-offs.
-
Deepen the GP-level analysis: extend GP analysis with practice-level characteristics (list size, type of organization, age/gender of GP). Otherwise, conclusions on GP irrelevance remain tentative.
-
Address COVID-19 bias explicitly: acknowledge and model the potential distorting effect of 2021 pandemic on ED use (psychiatric and social issues, reduced acute presentations). Could stratify findings pre-vs post-wave or compare to pre-pandemic baseline.
-
Strengthen SES analysis: provide stratified results combining SES and morbidity (e.g., multimorbidity among deprived vs affluent groups). Explore geography + SES interactions.
-
Improve discussion of interventions: expand policy implications, which structural reforms (case management, improved chronic care, socioeconomic mitigation measures) may best reduce FU? Draw from EU best practices.
-
Enrich limitations section: add mention of lack of proximity measures, exclusion of the homeless, retrospective bias, and dataset reliance on the main diagnosis.
Author Response
Comments 1: Clarify the FU definition rationale: justify why ≥ 4 was chosen over ≥3 or ≥5. Possibly add sensitivity analysis of risk factors across cut-offs.
Response 1: Thank you for your suggestion. The choice of threshold is based on the literature. This definition is present in most of the literature, both Italian and foreign. Other cut-offs were presented only in the description of the study population, while the most widely used cut-off in the literature (≥4) was adopted in the models in order to avoid a “fishing expedition” and to ensure robustness of the results, also considering that the characteristics of FUs defined with different cut-offs are similar. We have added a sentence in the introduction to complete the definition (line 49).
Comments 2: Deepen the GP-level analysis: extend GP analysis with practice-level characteristics (list size, type of organization, age/gender of GP). Otherwise, conclusions on GP irrelevance remain tentative.
Response 2: Thank you for your suggestion but we do not have information at general practitioner level. It would have been interesting to assess whether factors such as organizational affiliation, age, and gender might in some way influence the ability to manage a patient, but we also believe, from a statistical perspective, the model would likely have faced estimation issues, given that the analysis was conducted using a three-level model.
Comments 3: Address COVID-19 bias explicitly: acknowledge and model the potential distorting effect of 2021 pandemic on ED use (psychiatric and social issues, reduced acute presentations). Could stratify findings pre-vs post-wave or compare to pre-pandemic baseline.
Response 3: Thank you for pointing this out. Actually we do not believe that the results of our study are affected by the pandemic. All patients included in the analysis (those with at least one ED visit in 2021) were equally exposed to the pandemic waves, with no systematic differences between subgroups that could have introduced bias. Other studies conducted in the Lazio region compared ED visits during the COVID-19 waves and found a sharp reduction in ED visits, with the exception of those due to pneumonia (Pinnarelli L, Colais P, Mataloni F, Cascini S, Fusco D, Farchi S, Polo A, Lacalamita M, Spiga G, Ribaldi S, Magnanti M, Davoli M. Access to the Emergency Department in the time of COVID-19: an analysis of the first three months in the Lazio Region (Central Italy). Epidemiol Prev. 2020 Sep-Dec;44(5-6):359-366. Italian. doi: 10.19191/EP20.5-6.P359.011. PMID: 33706488). This type of assessment does not fall within the objectives of our study. We added a sentence in Discussion (lines 231-236).
Comments 4: Strengthen SES analysis: provide stratified results combining SES and morbidity (e.g., multimorbidity among deprived vs affluent groups). Explore geography + SES interactions.
Response 4: Thank you for your suggestion. Unfortunately, it was not possible to perform this type of analysis due to issues of stability and robustness of the results. The model we applied (a logistic model with a three-level hierarchical structure: patient - GP - District) does not allow the inclusion of additional covariates for further stratification or the addition of interaction terms without compromising the reliability of the estimates.
Comments 5: Improve discussion of interventions: expand policy implications, which structural reforms (case management, improved chronic care, socioeconomic mitigation measures) may best reduce FU? Draw from EU best practices.
Response 5: Thank you for your suggestion. We have added two systematic reviews, Italian and English, in order to describe two best practices in FU management. We extended the discussion (lines 267-279).
Comments 6: Enrich limitations section: add mention of lack of proximity measures, exclusion of the homeless, retrospective bias, and dataset reliance on the main diagnosis.
Response 6: The limitation section underlines the cited bias.
Reviewer 2 Report
Comments and Suggestions for Authors
- Please review and ensure that the keywords are selected and written according to the Medical Subject Headings (MeSH) available at the following address: https://www.ncbi.nlm.nih.gov/mesh"
- The introduction of the manuscript should clearly state the problem and the necessity of studying the topic. Although the authors have referred to useful points, the problematic nature of the subject is still unclear. Is the cost it imposes on the country the real issue? Is the fatigue of the healthcare staff the problem? Since it is repeatedly stated in the problem statement that frequent users are patients belonging to lower social classes, it seems there is no real problem. The issue arises only when the demands are induced rather than genuine.
- The explanations related to the study context in the last paragraph should be moved to the methodology section.
- Although the results are well presented, in the discussion section of the article, the results have not been adequately interpreted and analyzed. It is expected that the authors provide possible reasons to justify the findings. For example, it is generally assumed based on previous studies that women visit more frequently than men, but in the present study, the opposite is observed, and the reasons for this have not been explained.
- Practical and clear recommendations based on the results should be provided in a way that highlights the significance of the present study.
Author Response
Comments 1: Please review and ensure that the keywords are selected and written according to the Medical Subject Headings (MeSH) available at the following address: https://www.ncbi.nlm.nih.gov/mesh"
Response 1: Thank you for your comment. We have inserted the keywords using the Medical Subject Headings (MeSH) you suggested.
Comments 2: The introduction of the manuscript should clearly state the problem and the necessity of studying the topic. Although the authors have referred to useful points, the problematic nature of the subject is still unclear. Is the cost it imposes on the country the real issue? Is the fatigue of the healthcare staff the problem? Since it is repeatedly stated in the problem statement that frequent users are patients belonging to lower social classes, it seems there is no real problem. The issue arises only when the demands are induced rather than genuine.
Response 2: Thank you for your suggestion. We added a sentence in the Introduction to better clarify the consequences of inappropriate attendances and ED overcrowding (line 43).
Comments 3: The explanations related to the study context in the last paragraph should be moved to the methodology section.
Response 3: We agree. We moved the entire paragraph to the beginning of Methodology section.
Comments 4: Although the results are well presented, in the discussion section of the article, the results have not been adequately interpreted and analysed. It is expected that the authors provide possible reasons to justify the findings. For example, it is generally assumed based on previous studies that women visit more frequently than men, but in the present study, the opposite is observed, and the reasons for this have not been explained. Practical and clear recommendations based on the results should be provided in a way that highlights the significance of the present study.
Response 4: Thank you for your observation. We added two focus on the Covid19 pandemic (lines 231-236) and policies and care management (lines 267-279). We also added another hypothesis on the GP influence on FUs (line 257).
Round 2
Reviewer 1 Report
Comments and Suggestions for Authors
The revision did not address the limitations pointed on the previous round.
Author Response
Comment 1: The revision did not address the limitations pointed on the previous round
Response 1: We addressed many of the revisions required by reviewers without compromising the reliability of the estimates, and expanded the limitation section as suggested.
Reviewer 2 Report
Comments and Suggestions for Authors
Although the authors have addressed many of the revisions, concerns regarding the problem statement remain. The problem statement in the manuscript is still not well-structured. It is recommended that the problem statement be revised to follow the general principle of moving from a broad overview to specific details, ensuring a clear and coherent progression that convincingly demonstrates the necessity of the study to the readers
Author Response
Comment 1: Although the authors have addressed many of the revisions, concerns regarding the problem statement remain. The problem statement in the manuscript is still not well-structured. It is recommended that the problem statement be revised to follow the general principle of moving from a broad overview to specific details, ensuring a clear and coherent progression that convincingly demonstrates the necessity of the study to the readers
Response 1: Thank you for your comment. We've expanded the introduction to better explain the problem statement, emphasizing the importance and rationale for analyzing social and clinical characteristics.